# Aggregation Susceptibility of Low-Density Lipoproteins—A Novel Modifiable Biomarker of Cardiovascular Risk

**DOI:** 10.3390/jcm10081769

**Published:** 2021-04-19

**Authors:** Katariina Öörni, Petri T. Kovanen

**Affiliations:** 1Wihuri Research Institute, 00290 Helsinki, Finland; petri.kovanen@wri.fi; 2Molecular and Integrative Biosciences Research Programme, Faculty of Biological and Environmental Sciences, University of Helsinki, 00014 Helsinki, Finland

**Keywords:** foam cell, low-density lipoprotein, LDL aggregation, LDL retention, lipid accumulation

## Abstract

Circulating low-density lipoprotein (LDL) particles enter the arterial intima where they bind to the extracellular matrix and become modified by lipases, proteases, and oxidizing enzymes and agents. The modified LDL particles aggregate and fuse into larger matrix-bound lipid droplets and, upon generation of unesterified cholesterol, cholesterol crystals are also formed. Uptake of the aggregated/fused particles and cholesterol crystals by macrophages and smooth muscle cells induces their inflammatory activation and conversion into foam cells. In this review, we summarize the causes and consequences of LDL aggregation and describe the development and applications of an assay capable of determining the susceptibility of isolated LDL particles to aggregate when exposed to human recombinant sphingomyelinase enzyme ex vivo. Significant person-to-person differences in the aggregation susceptibility of LDL particles were observed, and such individual differences largely depended on particle lipid composition. The presence of aggregation-prone LDL in the circulation predicted future cardiovascular events in patients with atherosclerotic cardiovascular disease. We also discuss means capable of reducing LDL particles’ aggregation susceptibility that could potentially inhibit LDL aggregation in the arterial wall. Whether reductions in LDL aggregation susceptibility are associated with attenuated atherogenesis and a reduced risk of atherosclerotic cardiovascular diseases remains to be studied.

## 1. Introduction

Strong evidence from epidemiologic, genetic, and clinical studies has shown that low-density lipoprotein (LDL) particles are causal agents in atherogenesis, the process causing atherosclerotic cardiovascular disease (ASCVD) [1]. During atherogenesis, circulating LDL particles enter the subendothelial extracellular space of the arterial wall. In contrast to other peripheral tissues, in which the concentration of LDL particles in the extracellular fluid is only 10% of that in the circulating blood, in the arterial intima, the concentration of LDL particles is about the same as in the circulation—i.e., it is 10-fold higher than in other tissues [2]. The exceptionally high LDL concentration in the intima develops because the intima is largely a closed space for the LDL particles, as it is separated from the medial layer by a largely impermeable elastic lamina. Moreover, the extracellular fluid in the normal intima and the early developing lesions is not drained by lymphatic capillaries, which, like in the advancing aortic stenosis, appear to develop only in advanced atherosclerotic lesions [3,4,5].

Within the intimal space, a fraction of the LDL particles becomes entrapped by binding to the negatively charged components of the extracellular matrix [6,7]. The free-floating, and particularly the matrix-bound LDL particles become enzymatically modified by proteases and lipases, and also by pro-oxidative agents [8,9]. Morphologically, the initiation of atherosclerosis is characterized by an extracellular accumulation of small lipid droplets and vesicles that are associated with the extracellular matrix of the intima [10,11]. As reviewed by Öörni and coworkers [8], the analysis of the LDL-like particles isolated from rabbit and human atherosclerotic lesions showed signs of slight modification, whereas the lipid droplets and aggregated particles had features of extensive modification of LDL particles.

In a recent comprehensive analysis of the extracellular lipids accumulated in advanced human atherosclerotic plaques, ultrastructural 3D electron microscopical analysis of the plaques was combined with proteomic and lipidomic analysis of the extracellular lipids isolated from the plaques [12]. The visual observations demonstrated the presence of extracellularly located, aggregated and fused particles, and omics analyses of the isolated aggregates suggested that the particles had originated from lipase-modified apoB-containing circulating lipoprotein particles.

## 2. LDL Particles—The Major Cholesterol Carriers in the Circulation

LDL particles carry the majority of cholesterol in the circulating blood. They are the end products of the intravascular lipid-remodeling cascade of the apolipoprotein (apo) B-100-containing lipoproteins. LDL originates from the triglyceride-rich very-low-density lipoproteins (VLDLs), which the liver continuously synthesizes and secretes [13]. The gradual conversion of VLDLs into intermediary density lipoproteins and then into LDLs occurs when endothelial lipase hydrolyzes VLDL-triglycerides, with ensuing gradual depletion of them. For the formation of the end product, i.e., the LDLs, cholesteryl ester transfer protein activity is also of importance. It promotes the transfer of cholesteryl esters from high-density lipoproteins (HDLs) to the apoB-100-containing lipoproteins and, in exchange, triglycerides from them to the HDLs [14,15]. Accordingly, upon the lipoprotein conversion cascade, the mass percent composition of the apoB-100-containing lipoproteins changes, so that the content of triglycerides (TGs) decreases from about 60% in VLDLs to 10% in LDLs, and that of cholesteryl esters (CEs) increases from about 10% to 40% [16,17].

More specifically, the LDL particles have an average diameter of 22 nm, and they are composed of an amphipathic phospholipid surface layer and a hydrophobic lipid core [18]. Each LDL particle contains a single apoB-100 molecule, which is partially embedded in the surface lipid monolayer. The layer is composed of about 400 unesterified cholesterol (UC) molecules and 800 phospholipid molecules, of which about 500 are phosphatidylcholine (PC) and 200 sphingomyelin (SM) molecules. Moreover, the surface also contains about 80 lysophosphatidylcholine and a few phosphatidylethanolamine and ceramide molecules [19,20,21]. The lipid core of an LDL particle, in turn, contains an average of 1600 CE, 200 UC, and 170 TG molecules [18,22].

## 3. Potential Mechanisms of Aggregation of LDL Particles in the Arterial Intima

When LDL particles enter the arterial intima, they become exposed to oxidants and oxidizing enzymes, proteases, and lipases. For the aggregation of LDL particles, at least in vitro, modification of their surface structure is necessary [8]. Such modification can be achieved by various means, including physical force, such as vortexing [23]. Aggregated extracellular lipid droplets can also be generated from LDL in the arterial intima in vivo. Thus, aggregated lipid droplets were observed in the aortic intima of normal rabbits 2 h after an intravascular bolus injection of a large quantity of human LDLs [24]. Lipoproteins isolated from human atherosclerotic lesions vary in size from LDL-like particles to uni- and multilamellar vesicles, and ultimately to large lipid droplets (Table 1) [8]. The isolated particles are often aggregated, the aggregates varying from clusters of a few lipoproteins to micron-sized complexes, which may be connected to cholesterol crystals [12,21,25,26,27].

The particles isolated from atherosclerotic lesions show signs of multiple modifications, and several types of oxidative and enzymatic modifications of LDL particles in vitro can induce the formation of aggregated lipid droplets that resemble those seen in the lesions. Compared to plasma LDL particles, the particles isolated from atherosclerotic lesions have a lower PC and a higher lysophosphatidylcholine content [32,33], which suggests that their phospholipids have been oxidatively modified or hydrolyzed by phospholipase A_2_. Extensive oxidation modifies both apoB-100 and the lipid components of LDL and induces the formation of oxidation-specific epitopes on LDL particles [34]. Interestingly, very large LDL aggregates isolated from human atherosclerotic lesions were strongly (10- to 50-fold) enriched in ceramide, a product of SM cleavage by sphingomyelinase (SMase) [21].

Phospholipase A_2_ [35] and SMase [36] both induce changes in the conformation of apoB-100 and, at least in the case of SM hydrolysis, LDL aggregation is mediated via protein-protein interactions that occur after lipolysis-induced conformational changes in apoB-100 [36]. The lipid composition of LDL influences the conformation of apoB-100, and the high content of PC on the LDL surface appears to minimize lipolysis-induced changes in the protein conformation, and so stabilize the particles [37]. Regarding phospholipolysis, an interesting subfraction of LDL, electronegative LDL, contains phospholipolytic activity that can trigger aggregation of this LDL subfraction [38,39,40].

ApoB-100 in lipoproteins isolated from human atherosclerotic lesions is often fragmented [26]. This may be partly due to oxidation, but also due to the action of one or several proteases that are present in atherosclerotic lesions. These proteases include plasma-derived proteases and proteases secreted by inflammatory cells present in the lesions, particularly matrix metalloproteases and lysosomal proteases secreted by macrophages and neutral proteases secreted by activated mast cells [41,42,43,44,45,46,47]. All these proteolytic enzymes have been shown to induce aggregation and/or fusion of LDL particles in vitro, enhancing the interaction of the particles with the components of the extracellular matrix present in the arterial intima [8].

Proteolysis sensitizes LDL particles to other modifications, including hydrolysis of the core cholesteryl esters [12,48,49]. In fact, a large proportion of cholesterol in lipid particles accumulating in human atherosclerotic lesions is unesterified [31], which promotes cholesterol crystallization. Cholesterol crystals are frequently found in atherosclerotic lesions, and in advanced human carotid atherosclerotic plaques they are connected to aggregated lipoproteins [12].

Antibodies generated against LDL particles that have been modified with a combination of protease and cholesteryl esterase (called enzymatically modified LDL or E-LDL) detect lipid deposits in human atherosclerotic lesions [49]. Such double-modification induces LDL aggregation and modifies the surface of LDL to promote binding of complement components to the particles [50], a finding that connects modified and aggregated LDL to inflammation. The C-reactive protein can bind to modified LDL particles, particularly to unesterified cholesterol on the particle surface [51], and it has been shown to further enhance the binding of complement components to modified LDL particles [52,53].

The structural components and the varying pathophysiological conditions in the arterial intima likely influence the extent of LDL modification and aggregation within an evolving atherosclerotic plaque. Interaction of LDL with glycosaminoglycans induces structural changes in LDL particles that may promote their modification and change their aggregation behavior [54,55,56]. In the normal intima and early atherosclerotic lesions, the pH is neutral, thereby favoring the activity of enzymes with a neutral pH optimum, while advanced lesions can become hypoxic and acidic, thereby supporting the activity of enzymes with a more acidic pH optimum [57,58,59,60]. Many of the proteolytic and lipolytic enzymes present in atherosclerotic lesions have an acidic pH optimum and have, at least in vitro, the potential to extensively modify lipoprotein particles [43,45,61]. Accordingly, the enzymatic modifications of LDL particles are likely to be more diverse in advanced atherosclerotic lesions.

Modified LDL particles can also aggregate by binding to antibodies. In fact, most of the circulating oxidized LDL particles are bound to antibodies and found as oxLDL immune complexes [62], and the concentration of the circulating immune complexes is linked to atherosclerosis and cardiovascular events [63,64,65]. Although it is unlikely that such circulating immune complexes would easily cross the vascular endothelium and reach the arterial intima, immune complexes can be formed within the intima, where modified lipoproteins accumulate. Indeed, immune complexes are found in atherosclerotic lesions [66], where they can exert proinflammatory functions when taken up by macrophages.

## 4. Uptake of Aggregated LDL Particles with Ensuing Foam Cell Formation and Inflammasome Activation

Uptake of modified LDL particles by macrophages leads to foam cell formation, and several uptake mechanisms have been described depending on the type of LDL modification [67]. Thus, small LDL aggregates, such as those generated by, e.g., oxidation, have been reported to be taken up by macrophages via scavenger receptors. Modified lipoproteins bind, e.g., beta-amyloid, which opsonizes the lipoproteins and promotes their uptake [68]. Interestingly, large LDL aggregates induce in macrophages the generation of acidic extracellular surface-connected compartments into which the cells secrete lysosomal enzymes and protons to generate optimal conditions for the lysosomal enzymes [69,70]. This process requires actin polymerization and is mediated via Toll-like receptor 4 signaling [71,72]. LDL aggregates are partly hydrolyzed in the acidic compartments and their remnants are then internalized to be fully digested in the highly acidic lysosomes.

In the lysosomes, hydrolysis of cholesteryl esters leads to the generation of unesterified cholesterol, which can be re-esterified in the cytoplasm and be packed into lipid droplets typical of foam cells [73]. Uptake of oxidized LDL has been shown to lead in the lysosomes to the generation of cholesterol crystals [74], which can have significant proinflammatory consequences, as will be described below in more detail. Moreover, uptake of LDL aggregates generated by vortexing or by SMase treatment induces lysosomal accumulation of ceroid (lipofuscin), a product of lipid peroxidation consisting of complexes of protein and lipid [75]. Such lysosomal oxidation of LDL has been suggested to be iron-mediated and this oxidation can be inhibited by cysteamine [76].

Uptake of modified lipoproteins and lipids by the professional innate immune cells, such as the macrophages, can activate a multiprotein inflammatory complex termed the inflammasome [77]. Regarding atherosclerosis, the most studied inflammasome of the NLR family is the NLRP3 inflammasome. It requires priming and activation before secretion of the cytokines interleukin-1β and interleukin-18 takes place [78,79]. Expression of the components of the NLRP3 inflammasome is increased in atherosclerotic lesions [80]. Oxidized LDL, immune complexes, and cholesterol crystals have been shown to induce activation of the NLRP3 inflammasome [77]. Importantly, the addition of aggregated/fused lipid particles isolated from advanced atherosclerotic lesions to cultured human monocyte-derived macrophages can induce their conversion into foam cells and trigger inflammasome activation with ensuing IL-1β secretion [12]. Thus, the extracellular aggregated/fused lipoprotein particles have the potential to enhance atherogenesis in multiple ways, i.e., by contributing to extracellular lipid accumulation, intracellular cholesteryl ester accumulation, and local inflammation in an atherosclerotic plaque.

In addition to macrophages, smooth muscle cells (SMCs) can also take up modified lipoproteins and be converted into foam cells (reviewed recently in [81]). In fact, LDL aggregation and formation of LDL–proteoglycan complexes and LDL immune complexes appear to play major roles in forming SMC foam cells. SMCs can take up at least small LDL aggregates via the LDL receptor-related protein 1 (LRP-1) [82,83]. Aggregated LDL particles increase the release of soluble LRP-1 from cultured vascular smooth muscle cells, and also from human atherosclerotic plaques in an ex vivo model [84]. As a corollary, the soluble form of LRP-1 is detectable in the circulation, and its circulating levels are increased in patients with coronary artery disease [84,85].

## 5. Measurement of Aggregation Susceptibility of LDL Particles

We recently developed an assay to examine the aggregation susceptibility of LDL particles (Figure 1) [37,86]. In the assay, we isolate LDL particles from fresh or frozen plasma or serum samples and induce their aggregation with human recombinant SMase under conditions that promote the formation of micron-sized LDL aggregates [36]. LDL aggregation is then monitored by measuring the size of the forming LDL aggregates by dynamic light scattering. An LDL preparation that is prone to aggregate under these conditions has a strong tendency to aggregate. LDL particles aggregate also after their modification by oxidation, proteolysis, or phospholipolysis. However, the aggregation/fusion process induced by these other types of modification is slow and leads to the generation of LDL aggregates which are smaller than those after modification of LDL by the SMase [37].

## 6. Increased Aggregation Susceptibility of LDL Particles in Human Atherosclerotic Cardiovascular Disease—Particle Lipid Composition as an Explanatory Factor

We have shown that the aggregation susceptibility of LDLs is increased in patients with established atherosclerosis. Thus, patients with angiographically determined coronary stenosis had more aggregation-prone LDLs than healthy controls. Among the patients, those who died during the follow-up period had more aggregation-prone LDLs than patients with stable coronary artery disease, apparently reflecting unstable atherosclerotic plaques in the affected coronary arteries [37]. Similarly, patients with atherosclerotic occlusive peripheral arterial disease and who were undergoing lower extremity revascularization surgery had LDLs that aggregated much more rapidly than LDLs derived from healthy controls [87]. Importantly, the patients with the most aggregation-prone LDLs were more likely to experience a major acute cardiovascular event during the first year after the operation. More recently, we have shown that LDLs derived from children with familial hypercholesterolemia (FH) are more aggregation-prone than LDLs from healthy children [88]. However, the aggregation susceptibility of LDLs does not correlate with known risk factors for ASCVD, including serum LDL-cholesterol levels, age, gender, or the body mass index [37,87,88].

Ceramides are minor constituents of the circulating LDL particles. However, their amounts increase once the particles are retained in atherosclerotic lesions, where ceramides are found almost selectively in aggregated LDL particles due to of the activity of the SMase enzyme [21]. In this review, we have described the results obtained using a well-defined in vitro assay as a surrogate of the actual aggregation process in an atherosclerotic lesion’s complex metabolic environment. Thus, with the aid of a simple, surrogate two-component assay consisting of human plasma-derived LDL particles and SMase only, we could detect inter-individual differences in the susceptibility of circulating LDL particles to aggregate. Although we only studied the qualities of the SMase substrate (i.e., isolated LDL particles) without being able to measure the activity of the enzyme in the atherosclerotic coronary artery plaques, mere detection of the presence of highly aggregation-susceptible circulating LDL particles was able to predict future coronary events in patients with clinical coronary artery disease or peripheral artery disease.

Mechanistically plausible molecular connections exist between the quality of circulating LDL particles and a coronary event resulting from the rupture of an unstable coronary artery plaque. Such plaques are characterized by a large necrotic lipid core covered by a thin fibrous cap [77]. Interestingly, ceramide-containing macrophages are mainly found in such unstable human coronary atherosclerotic plaques [89]. Moreover, uptake of ceramide-containing LDL aggregates by macrophages induces the secretion of the matrix metalloproteinase-7 [37], an enzyme not only cleaving a wide array of extracellular matrix components present in the fibrous caps of an atherosclerotic lesion but also linked to actual plaque ruptures in humans [90].

Interestingly, the measurement of certain plasma ceramide species (as defined by their fatty acid content) and the ratios of such species can be used to predict major acute cardiovascular events and cardiovascular mortality [91,92,93,94,95]. In patients with known ASCVD, plasma ceramide levels may predict events even more accurately than the traditional risk biomarkers [96]. The multiple mechanisms potentially behind such observations include accumulation of ceramides in LDL particles which renders them susceptible to aggregate, and high proportions of ceramides and other sphingolipids joined by a low proportion of polyunsaturated PC species which further enhance the process of LDL aggregation [37]. Therefore, it is of interest that the predictive power of ceramides has recently been shown to improve if the ratios of ceramides and PC 22:5 and PC 22:6 (phosphatidylcholines with 22 carbons and five or six double bonds, respectively) are added to the prediction score [97,98].

The clinical significance of LDL-cholesterol is best exemplified in patients with FH having since birth plasma concentrations of LDL-cholesterol two to three times above the normal concentration. In these patients, atherosclerotic plaques and stenotic coronary artery disease may already start to develop when the patients are entering adulthood [99]. The clinical significance of LDL-cholesterol derives not only from its concentration but also the duration of the increased concentration [100]. When these two parameters, concentration and duration, are considered together, the cholesterol burden of the arterial wall, i.e., its cumulative exposure to the circulating LDL particles, can be calculated by multiplying LDL-cholesterol concentration by age in years [101]. Our recent demonstration of LDL’s increased aggregation susceptibility in children with FH [88] adds a third variable to the burden equation—the quality of LDL.

## 7. Attempts to Reduce the Aggregation Susceptibility of LDL Particles—From Molecular Understanding to Studies in Animal Models and Human Subjects

As discussed above, the aggregation susceptibility of LDL particles appears to largely depend on the lipid composition of LDL particles [37]. We have shown that a change in the lipid composition of LDL in vitro or in vivo in mice can change the aggregation susceptibility of the LDL particles (Figure 2). Thus, an increase in the PC content in vitro stabilized the LDL particles, while an increase in SM enhanced the SMase-induced LDL aggregation [37]. In accordance, an increase in LDL PC by injecting PC-vesicles into mouse circulation or a decrease in LDL SM by inhibition of SM biosynthesis in a mouse decreased LDL aggregation susceptibility [37]. We have also observed that LDL aggregation susceptibility can be modified in humans by diet or by medication [37,102]. Thus, increased consumption of vegetable oils and the addition of plant stanol-enriched spread were associated with reduced LDL aggregation. Moreover, a proprotein convertase subtilisin/kexin type 9 (PCSK9) inhibitor was found to reduce LDL aggregation [37]. In each case, the reduction in LDL aggregation susceptibility was strongly associated with an increase in the proportion of PC in the LDL particles [37,102,103].

LDL aggregation can also be inhibited by HDL particles, by the major protein component of such particles, apoA-I, or by other amphipathic α-helix-containing proteins [23,104]. We have previously shown that the apoA-I mimetic peptide 4F can also inhibit the SMase-induced LDL aggregation [105]. The 4F peptide was found to form an α-helical conformation upon binding to the surface of the SMase-modified LDL particles. The peptide was most likely able to stabilize the surface monolayer of LDL and so reduce particle aggregation. Notably, the 4F peptide has been shown to reduce experimental murine atherosclerosis, especially when applied during the early stages of atherosclerosis [106,107,108]. Recently, Rivas-Urbina et al. showed in LDL receptor-deficient mice that a 10-residue peptide from apoJ (clusterin) inhibits LDL aggregation by binding to modified LDL particles and that it also reduces atherosclerosis [109,110]. Thus, the LDL surface lipid monolayer-stabilizing effect of amphipathic peptides appears not to depend on the amino acid structure of the peptide, but rather on its ability to bind to the surface lipid monolayer. Both 4F and the apolipoprotein J-derived peptide are amphipathic and appear to achieve an α-helical conformation when bound to the LDL surface.

Finally, the therapeutic potential of HDL for atherosclerosis has been largely attributed to its major protein, apoA-I. Similar to apoA-I, the potent anti-atherogenic actions of the apoA-I mimetic peptide D-4F composed of D-amino acids, which are not degraded by gut peptidases and so allow its oral administration, largely rely on its anti-inflammatory activity and its ability to reduce oxidative stress [106]. Based on the results of several experimental studies presented in this review, HDL, apoA-I, and amphipathic peptides may directly or indirectly inhibit LDL aggregation, a propensity that could add to their proposed anti-atherogenic potential. As a protein, apoA-I is difficult and expensive to manufacture, and when used for therapeutic intervention it has required intravenous administration [101]. Interestingly, in early human studies in patients with a significant cardiovascular risk, a single dose of the oral D-4F rendered HDL less inflammatory, affirming its potential as a therapy to improve HDL function [111,112] and, perhaps, to also render LDL particles less aggregation-susceptible.

In the above context, the observation of an inverse relationship between the aggregation susceptibility of LDL and the cholesterol efflux-inducing ability of HDL is noteworthy [88]. However, preventive cardiovascular pharmacotherapies aimed at enhancing HDL quality and functionality have been repeatedly called into question [113]. Rather than aiming at HDL-dependent therapies, the current guidelines for the management of dyslipidemias in very high-risk patients call for an intensive triple-regimen (statin/ezetimibe/PCSK9 inhibitor) to effectively lower the concentration of LDL-cholesterol [114]. However, such dramatic lowering of LDL particle concentration may not equally reduce the concentrations of all subfractions of LDL particles [115,116]. Therefore, in patients with advanced atherosclerosis even after a dramatic reduction of LDL-cholesterol level, a residual risk remains [117]. Therefore, we still have room for supplementary approaches based on, e.g., therapeutic mitigation of the aggregation susceptibility of the remaining circulating LDL particles.

## 8. Conclusions and Future Directions

In this review, we addressed the causes and consequences of LDL aggregation. Particularly, we described an ex vivo assay that determines the aggregation susceptibility of circulating LDL particles. Such susceptibility shows high individual variation and appears to predict cardiovascular risk and to be amenable to manipulation. However, since the assay requires the isolation of LDL particles, it is cumbersome and so far not applicable for clinical use. Accordingly, further development and standardization of the assay are necessary before this biomarker may become clinically useful in detecting the risk of various atherosclerotic cardiovascular diseases.

## Figures and Tables

**Figure 1 jcm-10-01769-f001:**
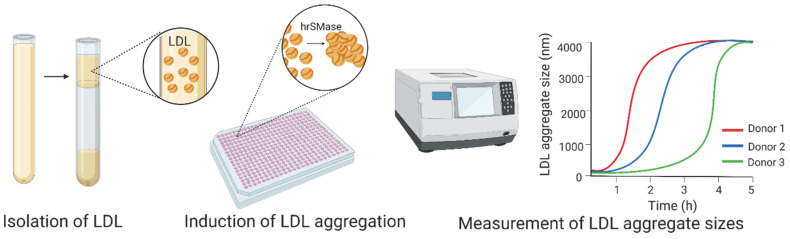
To determine low-density lipoprotein (LDL) aggregation susceptibility, LDL particles are first isolated by ultracentrifugation at a density of 1.019–1.063 g/mL. LDL aggregation is induced by the addition of human recombinant SMase, and LDL aggregation is then followed by monitoring the increase in LDL aggregate size by dynamic light scattering. LDL aggregation susceptibility is reported either as the average size of LDL aggregates at a certain time point or as the inflection point of the aggregate size vs. time curves [37,86]. hrSMase: human recombinant sphingomyelinase. Image created with BioRender.

**Figure 2 jcm-10-01769-f002:**
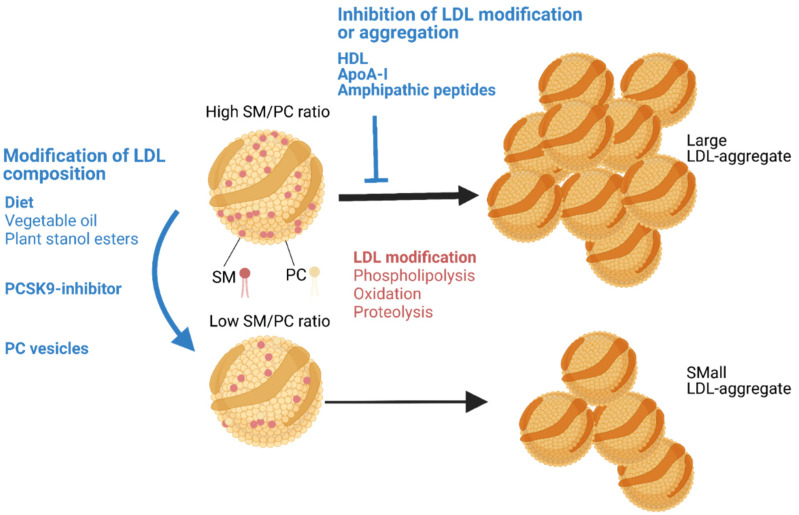
Aggregation LDL particles and its inhibition. LDL particles aggregate when they are modified by lipolytic or proteolytic enzymes or when they are exposed to pro-oxidative enzymes or agents. The aggregation susceptibility of LDL particles can be reduced (1) by modifying the lipid composition of the particles or (2) by inhibiting the modification of LDLs or stabilizing the modified LDL particles. The composition of circulating LDL particles can be modified by dietary changes and medications, such as PCSK9 inhibitors. These changes reduce the proportion of SM on the LDL particle surface. In a mouse model, injection of PC vesicles into the circulation has been shown to reduce the proportion of SM in LDL. Treatment of LDL particles in vitro by HDL, apoA-I, or amphipathic peptides can inhibit LDL modification or stabilize the surface of the modified LDL particles and directly inhibit their aggregation. Image created with BioRender.

**Table 1 jcm-10-01769-t001:** Sizes of lipoproteins and lipid particles isolated from human atherosclerotic lesions.

Lipoprotein	Size	Characteristics	Reference
Extracellular lipoproteins	20 to >200 nm	LDL-like particles and lipid droplets	[26,28,29,30]
Uni- and multilamellar vesicles	70 to 300 nm	Enriched in unesterified cholesterol	[30,31]
Aggregates	100 to 2000 nm	Connected to cholesterol crystals	[12,21]

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
