# Peer review of "Aggregation Susceptibility of Low-Density Lipoproteins—A Novel Modifiable Biomarker of Cardiovascular Risk"

_jcm, 2021, doi:10.3390/jcm10081769_

Round 1

Reviewer 1 Report

Overall:

Öörni and Kovanen have addressed an important issue on LDL aggregation and severity of atherosclerosis. The manuscript is very well written, justified questions and proper answers. Overall, the manuscript is timing and is beneficial to the field.

I have only following minor comments and suggestions:

-Based on the strong experience of the authors, could the authors add a table, which includes different LDL modification and corresponding particle size range?  

-Possibly this will be helpful if the authors also explain briefly “small dense LDL‐C” and “LDL‐C“and if there is a correlation between their size and severity of athresclorosis.

-As the authors are aware, dysfunctional HDL also could contribute atherosclerotic and cardiovascular disease. Would the authors include this type in the text?

-Since many scientists are working on modified LDL, they probably would try to correlate in vivo and in vitro LDL modification. If there is any data available, would the authors provide information how the scientist could modify LDL, which can reflect pathological LDL in the body (as close as possible).

-As far as I understood, LDL aggregation could be a prognosis factor for severity of the cardiovascular disease. However, the authors hesitated to call the aggregation factor as a marker. Would the author suggest and discuss whether measuring the aggradation susceptibility of LDL could be considered as a marker in the clinic?

-I think the paper needs a conclusion part, which provides a take home message.

 -Line 275, pleas correct “and bu also”     

Author Response

Response to the comments of Reviewer #1

We would like to thank the Reviewer for his/her comments and suggestions.

Comments and Suggestions for Authors

Overall:

Öörni and Kovanen have addressed an important issue on LDL aggregation and severity of atherosclerosis. The manuscript is very well written, justified questions and proper answers. Overall, the manuscript is timing and is beneficial to the field.

I have only following minor comments and suggestions:

-Based on the strong experience of the authors, could the authors add a table, which includes different LDL modification and corresponding particle size range?  

We have now added a table in which the sizes of lipoprotein particles and lipid droplets isolated from human atherosclerotic lesions are indicated. The sizes of LDL modified in vitro vary very much depending on the conditions used during their modification, so it would be difficult to add this information in a table.  

-Possibly this will be helpful if the authors also explain briefly “small dense LDL‐C” and “LDL‐C“and if there is a correlation between their size and severity of athresclorosis.

Small dense LDL has been shown to be more susceptible for oxidation, but we are not aware of studies examining LDL aggregation susceptibility of small dense LDL vs large buoyant LDL and, therefore, have not included this in the revised manuscript. On page 9 we shortly discuss different subfractions of LDL.  

-As the authors are aware, dysfunctional HDL also could contribute atherosclerotic and cardiovascular disease. Would the authors include this type in the text?

We have added information about the relationship of LDL aggregatibility and HDL function on page 8. We have kept discussion about HDL short due to a request from another reviewer.

-Since many scientists are working on modified LDL, they probably would try to correlate in vivo and in vitro LDL modification. If there is any data available, would the authors provide information how the scientist could modify LDL, which can reflect pathological LDL in the body (as close as possible).

Lipoprotein particles isolated from atherosclerotic lesions show signs of multiple modifications. We have modified the section 3 to more clearly indicate in which way each of the modifications change LDL particles. Unfortunately, there is no single modification that would entirely mimic lipoproteins isolated from the arterial wall.

-As far as I understood, LDL aggregation could be a prognosis factor for severity of the cardiovascular disease. However, the authors hesitated to call the aggregation factor as a marker. Would the author suggest and discuss whether measuring the aggradation susceptibility of LDL could be considered as a marker in the clinic?

Thank you for this comment. We have changed the title of the manuscript to reflect this idea and modified the final parts of the manuscript to include a more through discussion about the potential future use of the assay.

-I think the paper needs a conclusion part, which provides a take home message.

We have added a separate conclusion part into the manuscript. Thank you for the suggestion.

 -Line 275, pleas correct “and bu also”

We apologize for the typo. This has been corrected.

Reviewer 2 Report

General Comment: in this narrative review Öörni K. and Kovanen PT summarize the evidence around a new functional aspect of LDL particles, their aggregability, and the potential clinical relevance of its measurement ex vivo. The first part of the review summarizes in a nice and clear manner the pathophysiological mechanisms leading to LDL particles aggregation and the potential consequences of this process in the atherosclerotic disease. The second part of the review introduces the method of measurement, conceived by the Authors themselves, and reports some evidence of experimental studies performed by the Authors employing this technique. Although very interesting, and undoubtedly worth a narrative review, I found this second part too much self-promotional. The Authors do not report potential limitations for clinical application of their technique and they tend to overestimate the impact of their works (basically four papers, with a pre-print among them). These papers are indeed well-designed and really cutting-edge, but they can still be considered a sparse evidence.

Major comments:

  1. The title overestimates the relevance of the topic. There is no potential treatment specifically addressing LDL aggregability. Rather, the Authors have observed that some cholesterol-lowering treatments (namely plant sterols and PCSK-9 inhibitors) reduce the aggregability of LDL, likely through a modification of their lipid content. In this regard, Authors could also express an opinion on the expected effects of statin and apheresis on LDL aggregability. In my opinion, the most likely outlook in this topic is to refine the prediction of cardiovascular risk.
  2. Authors should report and discuss also the potential limitations of the technique they employed to measure aggregability. For instance, the technique measures the trend of LDL particles towards aggregation when exposed to a maximal stress. This is a useful information, but it does not necessarily correspond to what happens in tissues, where other factors could play a role.
  3. The paragraph about conclusion is too long and does not state any "true" conclusion. Most of the topics included in the conclusions (e.g. role of ceramides and Apo-AI) could be dealt in the previous paragraph, whereas the conclusion should be made of few rows, clearly declaring the take-home message.
  4. The Authors should downgrade the impact of the topic in clinical practice, clearly stating what is the state-of-the art and what is the personal opinion of the Authors. Rather should they highlight the novelty of the topic.

Minor comments:

Page 2, Lines 45-48 - this paragraph is unclear. What do the Authors mean for "LDL-like particles"? Where are the droplets and aggregates derived from? Why did the Authors specify "plasma-derived LDL" in this context?

Page 7, Lines 300-307 - the relevance of increasing the circulating levels of HDL, as well as the use of Apo-AI analogues, has been severely downgraded in the last years (see for instance 2018 ESC Guidelines for the treatment of hypercholesterolemia). 

Reviewer 3 Report

The present review provides interesting pathophysiological thought-provoking impulses concerning aggregation of low-density lipoproteins. The review is well written and comprehensive. However, there are main issues, which have to be addressed:

Major points:

„3. Potential mechanisms of aggregation of LDL particles in the arterial intima“

  1. A well-balanced paragraph on aggregation of LDL should not only recognize immunohistochemical detection of eLDL but also isolation and in vitro generation (1) of this modification, which provides a prime example of the author´s line of argument.
  2. Line 110 and 111: Reference (2) should be added.
  3. There is no mention made of possible interactions of LDL modifications and other molecules like CRP, beta-amyloid, e. t. c. and their impact on aggregation within the atherosclerotic lesion.

Minor points

Line 275: lesion and bu also?

References

  1. Bhakdi, S., B. Dorweiler, et al. (1995). "On the pathogenesis of atherosclerosis: enzymatic transformation of human low density lipoprotein to an atherogenic moiety." J Exp Med 182(6): 1959-1971.
  2. Torzewski, M., P. Suriyaphol, et al. (2004). "Enzymatic modification of low-density lipoprotein in the arterial wall: a new role for plasmin and matrix metalloproteinases in atherogenesis." Arterioscler Thromb Vasc Biol 24(11): 2130-2136.

Round 2

Reviewer 2 Report

All my comments have been adequately addressed. The quality level of the paper was already high before revision, but I thin it has further improved hereafter. A thorough language revision is suggested.

Reviewer 3 Report

The authors have addressed the main issues properly, I have no further comments.